# Sulfur Vacancies Enriched 2D ZnIn$_2$S$_4$ Nanosheets for Improving Photoelectrochemical Performance

**Sujuan Hu** [1,*], **Li Jin** [1], **Wangyu Si** [1], **Baoling Wang** [1] **and Mingshan Zhu** [2]

[1] School of Chemistry and Chemical Engineering, Kunming University, Kunming 650214, China; wujinjinli@163.com (L.J.); herryman@163.com (W.S.); healthygreenw@163.com (B.W.)

[2] School of Environment, Jinan University, Guangzhou 510632, China; zhumingshan@jnu.edu.cn

\* Correspondence: sujuanhu@kmu.edu.cn

**Abstract:** Vacancies engineering based on semiconductors is an effective method to enhance photoelectrochemical activity. Herein, we used a facile one-step solvothermal method to prepare sulfur vacancies modified ultrathin two-dimensional (2D) ZnIn$_2$S$_4$ nanosheets. The photon-to-current efficiency of sulfur vacancies modified ultrathin 2D ZnIn$_2$S$_4$ nanosheets is 1.82-fold than ZnIn$_2$S$_4$ nanosheets without sulfur vacancies and 2.04-fold than multilayer ZnIn$_2$S$_4$. The better performances can be attributed to the introduced sulfur vacancies in ZnIn$_2$S$_4$, which influence the electronic structure of ZnIn$_2$S$_4$ to absorb more visible light and act as the electrons trapping sites to suppress the recombination of photo-generated carriers. These results provide a new route to designing efficient photocatalyst by introducing sulfur vacancies.

**Keywords:** ZnIn$_2$S$_4$ nanosheets; sulfur vacancies; vacancies engineering; photoelectrocatalysts; photoelectrochemical performance





## 1. Introduction

Solar fuel produced by solar-driven photocatalysts has been regarded as one kind of latent low-cost renewable fuel that could replace fossil fuels. However, photoelectrochemical efficiency still faces huge challenges for practical applications [1–4]. Hopefully, the spring up of two-dimension materials brings up the potential that enhances photoelectrochemical performance due to the large specific surface areas, rich surface active sites, and nice photo-generated carrier separation rate [5–7]. Among numerous 2D semiconductor materials, 2D ZnIn$_2$S$_4$ nanosheets with their atoms arraying according to S-In-S-In-S-Zn-S gain more interest than their bulk structure because they have peculiar electronic structures, larger specific surface area, and shorter diffusion distance of charge carriers that would favor to the improvement of photoelectrochemical efficiency [8–10]. However, similar to other 2D semiconductors, carrier recombination is severe in 2D ZnIn$_2$S$_4$, which limits its photoelectrochemical performance.

Vacancy engineering is considered a convenient, potentially scalable, and effective strategy to improve photoelectrochemical performance. In addition to suppress the recombination of carriers of semiconductors, vacancy engineering can also change the electronic structure, optimize electrical conductivity, and regulate the active sites [11–14]. For example, surface oxygen vacancies were introduced in ferroelectric Bi$_3$TiNbO$_9$ nanosheets that could extend the photo-absorption of Bi$_3$TiNbO$_9$ nanosheets, promote the adsorption and activation of CO$_2$ molecules on the surface of the catalyst and maintain superb ferroelectric polarization for enhancing CO$_2$ photoreduction performance [15]. Similar advantages have been reported in WO$_3$ and TiO$_2$ with oxygen vacancies, as well as NiCo$_2$S$_4$ with sulfur vacancies [16–18]. Therefore, sulfur vacancies designed in the 2D ZnIn$_2$S$_4$ nanosheets can highly probably promote photoelectrochemical performances [10,19].

Herein, ultrathin 2D ZnIn$_2$S$_4$ nanosheets modified with sulfur vacancies were prepared by a facile one-step solvothermal method. The ultrathin 2D ZnIn$_2$S$_4$ nanosheets modified

with sulfur vacancies exhibit higher photoelectrochemical performance than 2D $ZnIn_2S_4$ without sulfur vacancies and multilayer $ZnIn_2S_4$. The introduced sulfur vacancies in ultra-thin 2D $ZnIn_2S_4$ nanosheets effectively strengthen visible light absorption and inhibit the recombination of photogenerated carriers, synergistically promoting photoelectrochemical performances.

## 2. Results and Discussion

XRD patterns of 2D ZIS, 2D ZIS-$S_v$, and multilayer 2D ZIS are shown in Figure 1A. The diffraction peaks are well indexed to a hexagonal $ZnIn_2S_4$ (JCPDS NO. 65-2023) [19]. Additionally, the $2\theta$ diffraction peaks located at 21.7°, 27.6°, and 47.4° are assigned to the (006), (102), and (110) planes, respectively. Due to the introduction of abundant TAA in the synthesis process, the peak at 30.1° became weaker. The morphology of 2D ZIS-$S_v$ is a thin nanosheet (Figure 1D,G), the tagged interplanar crystal spacing of 0.32 nm can be ascribed to the $ZnIn_2S_4$ (102) plane and some vacancies and the lattice distortion can be observed (inset of Figure 1B) [19]. Further, the homogeneous distribution of the Zn, In, and S elements in sulfur vacancies modified $ZnIn_2S_4$ are observed by the HRTEM mapping scan (Figure 1C,E,F,H,I). The morphology of ZIS-$S_v$ is a thin nanosheet (Figure 1G) and the tagged interplanar crystal spacing of 0.32 nm can be ascribed to the $ZnIn_2S_4$ (102) plane (inset of Figure 1B) [19]. XPS measurements were conducted to further investigate the presence of vacancies in $ZnIn_2S_4$ (Figure 2). The high-resolution XPS spectra of S 2p in all specimens can be fitted to two peaks that are assigned to divalent sulfide ions ($S^{2-}$). The binding energies of S $2p_{3/2}$ (161.63 eV) and S $2p_{1/2}$ (162.88 eV) in 2D ZIS-$S_v$ are negative and broader than those of 2D ZIS (161.90 eV and 163.04 eV) and multilayer 2D ZIS (161.94 eV and 163.09 eV), indicating that sulfur vacancies exist because of low-coordination sulfur [20]. The binding energies of Zn $2p_{3/2}$ (1021.56 eV) and S $2p_{1/2}$ (1044.71 eV) are negative shifted ca. 0.40 eV than multilayer 2D ZIS (1021.99 eV and 1045.07 eV), however, this phenomenon is not obvious by In 3d, suggesting that the loss of sulfur atoms occurs around Zn atoms instead of In atoms [20]. To further prove the existence of sulfur vacancies in the samples, EPR was measured, as shown in Figure 3A. An obvious resonance signal at g = 2.003 was detected only in 2D ZIS-$S_v$ and this signal resulted from the existence of unpaired electrons of vacancies [21]. Through the above analysis, it can be inferred that abundant sulfur vacancies exist in 2D ZIS-$S_v$. Spectral properties of 2D ZIS, 2D ZIS-$S_v$, and multilayer 2D ZIS were measured by UV-vis DRS, as shown in Figure 3B. In comparison with multilayer 2D ZIS (Eg≈2.24 eV) and 2D ZIS (Eg≈2.20 eV), 2D ZIS-$S_v$ shows a narrower band-gap (ca. 2.12 eV) because of the existence of sulfur vacancies that help to absorb more visible light.

Photoelectrochemical measurements were conducted for investigating the performances [22,23]. Figure 4A–C are the linear sweep voltammetry for 2D ZIS, 2D ZIS-$S_v$, and multilayer 2D ZIS with and without light irradiation to evaluate the onset potential. It can be seen that the photocurrent densities for 2D ZIS, 2D ZIS-$S_v$, and multilayer 2D ZIS in dark are slight. However, when exposed to light irradiation, 2D ZIS, 2D ZIS-$S_v$, and multilayer 2D ZIS exhibit apparent enhancement with photocurrent densities as high as 79.5, 122.8, 77.2 µA mg$^{-1}$ at 0.3 V vs. SCE, respectively, and 2D ZIS-$S_v$ exhibits the highest improvement. Further, all the samples exhibit lower onset potential values under light irradiation (here we use the intersection point potential between the dark current and the tangent at the maximum slope of photocurrent to denote the onset potential) than those in the dark. Based on LSV results, the photoelectrochemical efficiencies of all the samples can be evaluated quantitatively utilizing applied bias photon-to-current efficiency (ABPE) as the formula below [24]:

$$ABPE = \left[ \frac{J_{ph}(1.23 - V_b)}{P_{total}} \right] \times 100\%$$

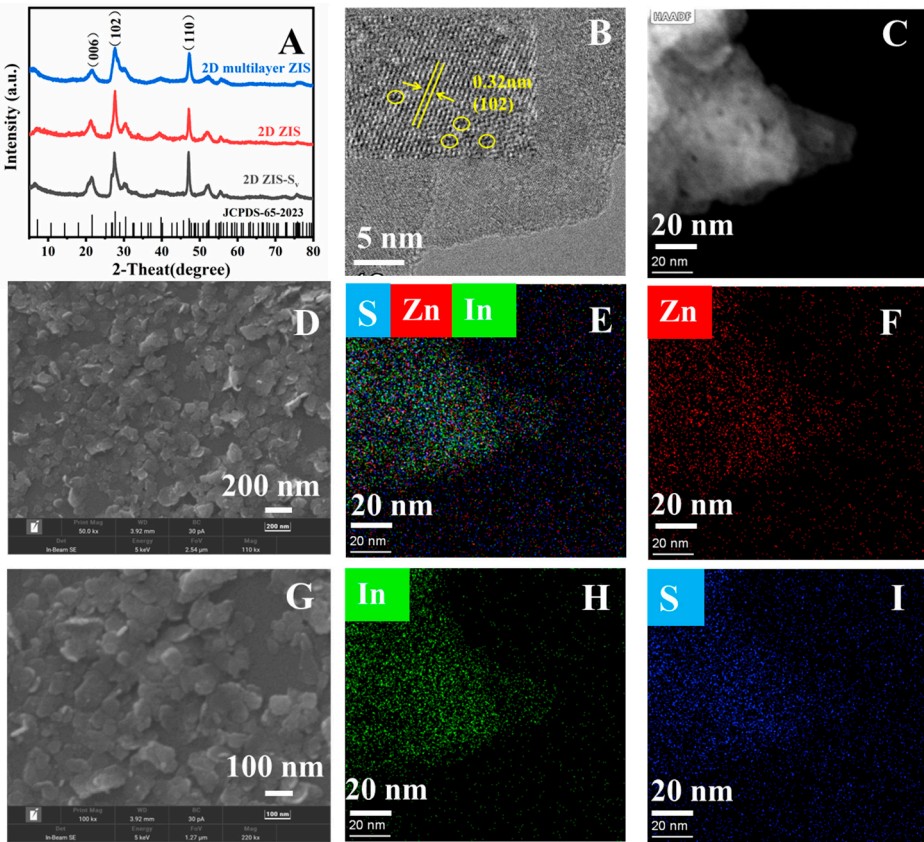

**Figure 1.** XRD patterns (**A**), SEM (**D,G**), HRTEM (**B**), and the STEM (**C**), elemental mappings (**E**) of Zn (**F**), In (**H**) and S (**I**) of 2D ZIS-S$_v$.

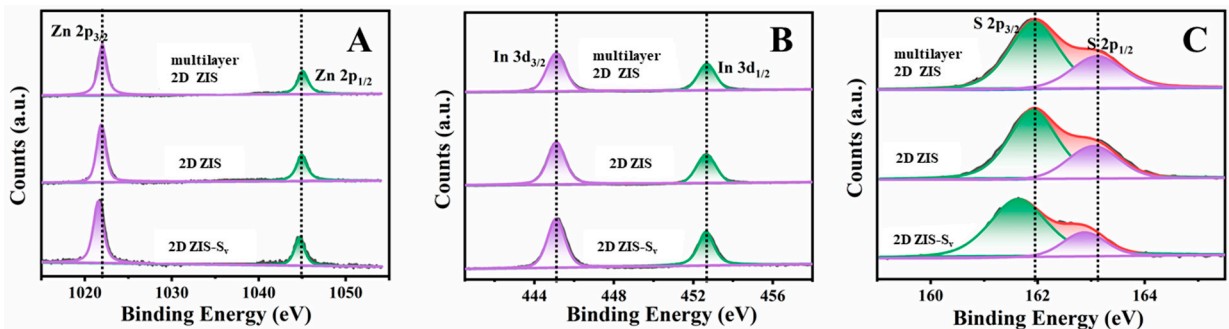

**Figure 2.** High-resolution XPS spectra for Zn (**A**), In (**B**), and S (**C**) of 2D ZIS, 2D ZIS-S$_v$, and multilayer 2D ZIS.

$J_{ph}$ denotes the photocurrent density of scanning potential, $V_b$ means the applied bias, and $P_{total}$ is the light power density. The calculated ABPE efficiencies of 2D ZIS, 2D ZIS-S$_v$, and multilayer 2D ZIS under light irradiation are 0.076%, 0.139%, and 0.068%, respectively (Figure 4D–F). The photon-to-current efficiency of sulfur vacancies modified ultrathin 2D ZnIn$_2$S$_4$ nanosheets is 1.82-fold than 2D ZnIn$_2$S$_4$ nanosheets without sulfur vacancies and 2.04-fold more than multilayer 2D ZnIn$_2$S$_4$.

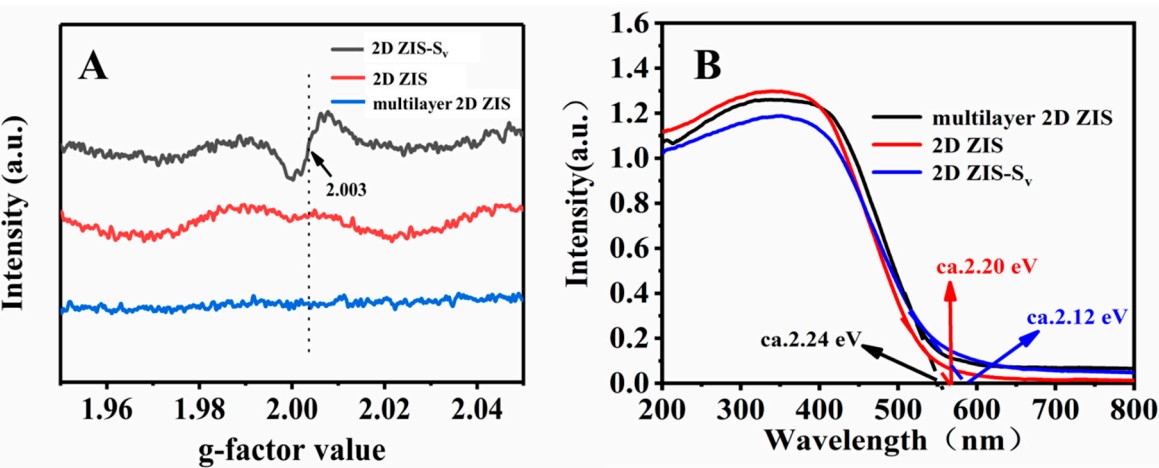

**Figure 3.** EPR (**A**) and UV-vis DRS (**B**) of 2D ZIS, 2D ZIS-S$_v$, and multilayer 2D ZIS.

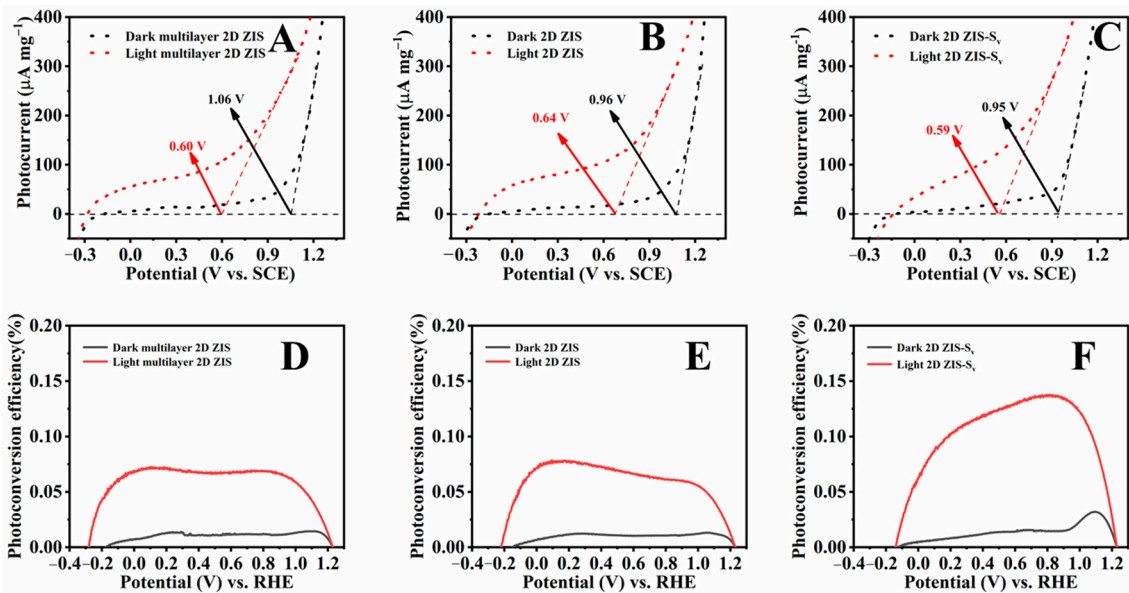

**Figure 4.** Linear sweep voltammetry curves of multilayer 2D ZIS (**A**), 2D ZIS (**B**), and 2D ZIS-S$_v$ (**C**) under light irradiation and in dark; photoconversion efficiency of multilayer 2D ZIS (**D**), 2D ZIS (**E**), and 2D ZIS-S$_v$ (**F**) under light irradiation and in dark.

The electrochemical impedance spectra (EIS) for 2D ZIS, 2D ZIS-S$_v$, and multilayer 2D ZIS are demonstrated in Figure 5. All the samples show smaller diameters under light irradiation than those in the dark, indicating a smaller resistance and faster interfacial electron transfer under light irradiation. Figure 6 shows transient photocurrent density curves for 2D ZIS, 2D ZIS-S$_v$, and multilayer 2D ZIS. All samples exhibit rapid and stable photocurrent responses. In addition, the value of transient photocurrent density for 2D ZIS-S$_v$ is ca. six times higher than that of multilayer 2D ZIS, suggesting that 2D ZIS-S$_v$ has a better separation of photo-generated carriers and a longer lifetime of photo-generated carriers [25,26]. For verifying this, PL spectra and TRPL decay spectra were performed, as shown in Figure 7A,B. All the samples exhibit the strongest PL peak at ca. 545 nm, which originates from the photogenerated carriers' recombination from band-gap transition. Among the three samples, 2D ZIS-S$_v$ exhibits the weakest PL peak due to the sulfur vacancies effectively inhibiting the recombination of photo-generated carriers and producing more electrons and holes for photocatalysis. The photo-generated carrier lifetime

can be measured by TRPL spectra in Figure 7B. The average lifetime ($\tau_A$) can be calculated on the basis of the equation [27]:

$$\tau_A = \frac{A_1\tau_1{}^2 + A_2\tau_2{}^2}{A_1\tau_1 + A_2\tau_2}$$

where $\tau_1$ and $\tau_2$ are the fluorescent lifetime, $A_1$ and $A_2$ are the corresponding amplitudes. According to the fitting data $\tau_A$, 2D ZIS shows a shorter lifetime of 4.98 ns and 2D ZIS-S$_v$ exhibits a longer lifetime of 6.89 ns, indicating that introducing sulfur vacancies can extend the lifetime of carriers [28,29].

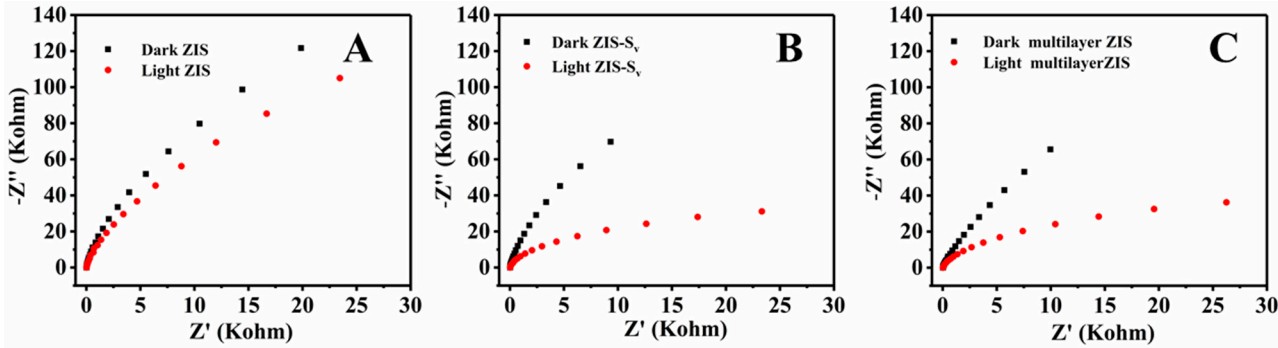

**Figure 5.** EIS of 2D ZIS (**A**), 2D ZIS-S$_v$ (**B**), and multilayer 2D ZIS (**C**).

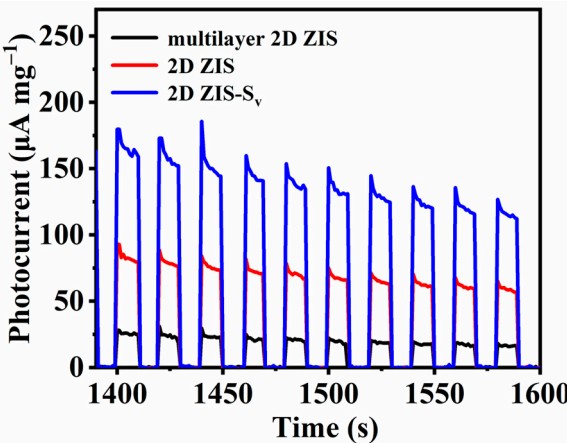

**Figure 6.** Photocurrent responses of 2D ZIS, 2D ZIS-S$_v$, and multilayer 2D ZIS.

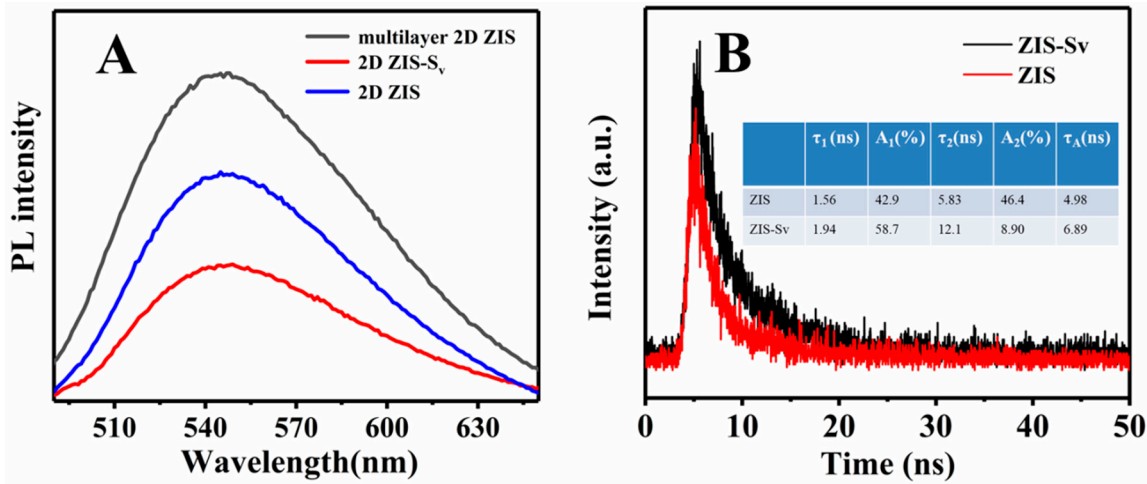

| | $\tau_1$ (ns) | $A_1$(%) | $\tau_2$(ns) | $A_2$(%) | $\tau_A$(ns) |
|---|---|---|---|---|---|
| ZIS | 1.56 | 42.9 | 5.83 | 46.4 | 4.98 |
| ZIS-Sv | 1.94 | 58.7 | 12.1 | 8.90 | 6.89 |

**Figure 7.** PL spectra (**A**) and TRPL decay spectra (**B**) of samples.

Based on the above experimental results, 2D ZIS-$S_v$ exhibits the best photoelectrochemical activities. This may result from the introduction of sulfur vacancies in 2D ZIS, which influence the electron's structure of 2D ZIS that would help absorb more visible light. Further, sulfur vacancies can serve as the electron trapping center and suppress the recombination of photo-generated carriers. We use VB XPS spectra to obtain the valence band (VB) potentials of 2D ZIS, 2D ZIS-$S_v$, and multilayer 2D ZIS (Figure 8). Values 1.75, 1.38, and 1.69 eV were measured for 2D ZIS, 2D ZIS-$S_v$, and multilayer 2D ZIS, respectively. Therefore, 1.51, 1.14 and 1.45 eV (vs. NHE) were estimated for 2D ZIS, 2D ZIS-$S_v$ and multilayer 2D ZIS, respectively (according to formula $E_{VB, NHE} = \varphi + E_{VB, XPS} - 4.44$) [30]. The conductor band (CB) potentials were estimated to be −0.69, −0.98, and −0.75 eV for 2D ZIS, 2D ZIS-$S_v$, and multilayer 2D ZIS, respectively [31]. A possible mechanism based on the band structures is proposed in Figure 9. It is clearly seen that the sulfur vacancies change the CB and VB potentials and that the changed CB potential is more negative than others and $H^+/H_2$ (0 eV). Sulfur vacancies influence the electron's density of 2D ZIS and elevate the position of CB, allowing photo-excited electrons with a stronger reducing ability to take part in the photoelectrochemical process. Further, due to the reduction of thickness in 2D ZIS, the photogenerated electrons and holes can quickly, effectively, and without a barrier, reach the material surface and take part in redox reactions. The introduced sulfur vacancies in 2D ZIS serve as electron trapping centers which favor transferring the electrons from CB to sulfur vacancies and benefit the separation of electrons and holes. After that, the separated electrons reduce $H^+$ to $H_2$ in the cathode and the holes oxidize $H_2O$ to $O_2$ in the anode.

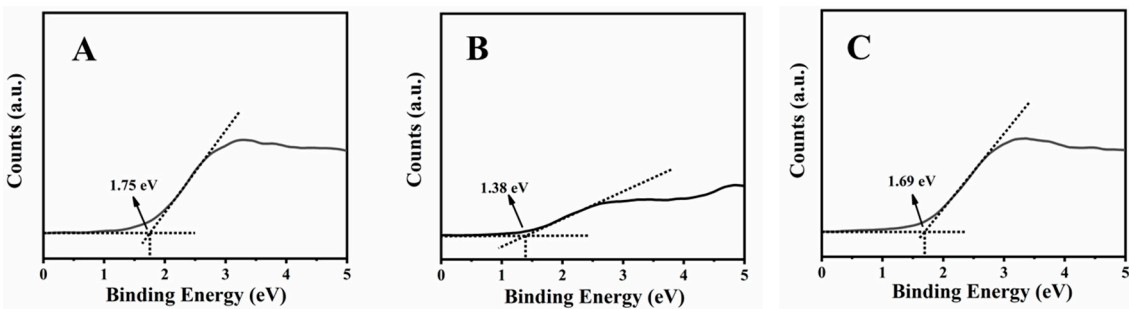

**Figure 8.** XPS valences of 2D ZIS (**A**), 2D ZIS-$S_v$ (**B**), and multilayer 2D ZIS (**C**).

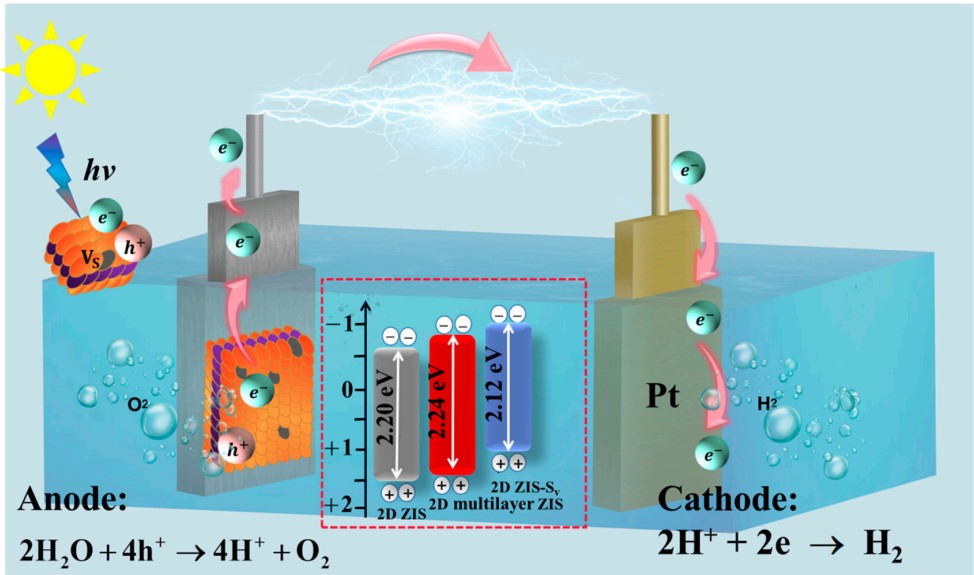

**Figure 9.** Mechanism illustration of photoelectrochemical water splitting of 2D ZIS-$S_v$ and the band structure of 2D ZIS, 2D ZIS-$S_v$, and multilayer 2D ZIS.

## 3. Experimental Section

All raw materials were purchased from Shanghai Aladdin Bio-Chemical Technology Co., Ltd. (Shanghai, China) without further purification. Deionized water was used throughout the experiments.

### 3.1. Synthesis of 2D Ultrathin $ZnIn_2S_4$ Nanosheets with and without Sulfur Vacancies and Multilayer $ZnIn_2S_4$

Herein, we take a one-step solvothermal method to fabricate 2D ultrathin $ZnIn_2S_4$ nanosheets with and without sulfur vacancies [20]. First, 1.6 mmol $InCl_3$, 0.8 mmol of $Zn(CH_3COO)_2 \cdot 2H_2O$, and 6.4 mmol thioacetamide (TAA) were added into a mixture containing 30 mL deionized water and 30 mL absolute ethanol with continuous stirring for 0.5 h. Then, the mixture was transferred into a 100 mL Teflon-lined stainless-steel autoclave and kept at 190 °C for one day. The precipitate, labeled as ZIS-$S_v$, was taken out after the Teflon-lined stainless-steel autoclave cooled down to room temperature and then washed with ethanol and water several times, respectively. Finally, the sample was achieved after being dried at 80 °C for 10 h in a stove. The process of the $ZnIn_2S_4$ without sulfur vacancies, labeled as 2D ZIS, is similar to that of the 2D ZIS-$S_v$, except the dosage of TAA is 3.2 mmol. For comparison, multilayer 2D $ZnIn_2S_4$ (multilayer 2D ZIS) was synthesized using a similar method of preparing 2D ZIS except that 60 mL of deionized water was used as a solvent.

### 3.2. Characterization

The samples were characterized by X-ray diffraction (XRD, Cu K$\alpha$ radiation, XPert), Scanning electron microscope (SEM, JSM-IT100), high-resolution transmission electron microscope (HRTEM, Talos-F200S), UV-visible diffuse reflectance spectroscopy (UV-DRS, Solid 2450), X-ray photoelectron spectroscopy (XPS, ESCALAB 250), fluorescence spectrophotometer (PL, FLS1000). Time-resolved photoluminescence spectroscopy (TRPL, FLS1000) was recorded under the excitation of a hydrogen flash lamp with 325 nm laser.

### 3.3. Photo-Electrochemical Measurement

First, the electrode was prepared as follows [32–34]: 1 mg sample, 0.5 mL ethanol, 0.5 mL deionized water, and 2 μL Nafion solutions (5 wt%) were well mixed and ultrasonically treated for 0.5 h. Then, the suspension (10 μL) was dropped onto the polished L-type glassy carbon electrode (effective area: 0.196 cm$^2$) and dried at room temperature.

Photoelectrochemical measurements were carried out on an electrochemical workstation (Autolab PGSTAT 302 N) with a three-electrode system, where Pt plate acted as the counter electrode, a saturated calomel electrode (SCE) acted as the reference electrode and a modified L-type GCE acted as the working electrode [35–37]. Light was produced by a xenon lamp (300 W PLS-SXE300C).

## 4. Conclusions

In this work, ultrathin 2D $ZnIn_2S_4$ nanosheets modified with sulfur vacancies were prepared by a facile one-step solvothermal method. The introduced sulfur vacancies in ultrathin 2D $ZnIn_2S_4$ nanosheets can effectively strengthen visible light absorption, inhibit the recombination of photogenerated carriers and extend the lifetime of photo-generated carriers. The ultrathin 2D $ZnIn_2S_4$ nanosheets modified with sulfur vacancies exhibit higher photon-to-current efficiency. Overall, vacancy engineering based on 2D semiconductor materials is an effective strategy to improve photoelectrochemical performance.

**Author Contributions:** Conceptualization, S.H.; methodology, S.H.; software, S.H. and L.J.; validation, S.H. and M.Z.; formal analysis, L.J.; investigation, L.J. and W.S.; resources, S.H. and B.W.; data curation, S.H. and L.J.; writing—original draft preparation, S.H.; writing—review and editing, S.H.; visualization, S.H.; supervision, M.Z.; project administration, S.H. and B.W.; funding acquisition, S.H. All authors have read and agreed to the published version of the manuscript.

**Funding:** This research was funded by the [National Nature Science Foundation of China] grant number [62004085], the [Special Basic Cooperative Research Programs of Yunnan Provincial Undergraduate Universities' Association] grant number [202101BA070001-034] ["Thousand Talents Program" of Yunnan Province for Young Talents] and [Innovative Research Teams (in Science and Technology) in the University of Yunnan Province (IRTSTYN)].

**Conflicts of Interest:** The authors declare no conflict of interest.

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
