# Peer review of "Sulfur Vacancies Enriched 2D ZnIn2S4 Nanosheets for Improving Photoelectrochemical Performance"

_catalysts, doi:10.3390/catal12040400_

Round 1

Reviewer 1 Report

The major novelty is proposed by S. Hu et al., herein is about the role the point defects i.e. (hopefully surface) S-atom vacancies of the 2D nanosheets ZIS (ZnIn2S4) as prepared from their exp. methodology. More specifically, the work provides not only the novel thoughts on the role of the surface S vacancies for the possible visible-light active photoelectrochemical (PEC) performance enhancement by the photocatalysts 2D-ZIS, but also a relative understanding on the PEC results compared to the multilayer counterpart of ZIS is also proposed. I feel that overall manuscript is well novel and focused into the direction of using two-dimensional (2D) nanomaterials for the solar energy applications.

Some major issues are present in the current manuscript, which must be revised before publication in the journal catalysts, which are as follows:

  1. Let us call the current ZIS as the 2D-ZIS and so on for the others, since the reader must have feelings at the beginning that it is a 2D materials different from the bulk ZIS.
  2. The results part should start from the discussion on the geometry i.e. combining the XRD, XPS and EPR results. Showing band gap from UV-vis-DRS is too first. However, on the EPR results the from 2D-ZIS and 2D-ZIS-Sv are markedly different than the literatures Refs. Journal of Energy Chemistry 58, 408-414 (2021) and ChemCatChem 13, 5148-5155 (2021). Thus, additional discussion is required by keeping in mind the difference of the 2D and bulk nature of the ZIS.
  3. Tauc plotted on DRS for band gap values must be correlated with the PL data, which could be the next subsection, while I do not see very clear correlation. Thus, common reader would have difficulty to understand the band gap evolution in few and multi layers of 2D-ZIS and or 2D-ZIS-Vs
  4. It will much interesting to see the HRTEM even at 5 nm scale (20 nm is too high). Since, vacancies are very prone to present also in the multi-layer ZIS, a rational discussion must be emphasized that how defects and 2D nature of ZIS is important such 2-fold increase of the PEC performance. I do not know if we will call it as a surface S-vacancy or sub-layer vacancies induced surface reconstruction in presence of the transition metals (Zn) ? This must be justified.

Author Response

Reviewer 1

The major novelty is proposed by S. Hu et al., herein is about the role the point defects i.e. (hopefully surface) S-atom vacancies of the 2D nanosheets ZIS (ZnIn2S4) as prepared from their exp. methodology. More specifically, the work provides not only the novel thoughts on the role of the surface S vacancies for the possible visible-light active photoelectrochemical (PEC) performance enhancement by the photocatalysts 2D-ZIS, but also a relative understanding on the PEC results compared to the multilayer counterpart of ZIS is also proposed. I feel that overall manuscript is well novel and focused into the direction of using two-dimensional (2D) nanomaterials for the solar energy applications.

 Some major issues are present in the current manuscript, which must be revised before publication in the journal catalysts, which are as follows:

  1. Let us call the current ZIS as the 2D-ZIS and so on for the others, since the reader must have feelings at the beginning that it is a 2D materials different from the bulk ZIS.

Reply: Thanks for the reviewer’s suggestions. The sulfur vacancies modified ultrathin 2D ZnIn2S4 nanosheets, 2D ZnIn2S4 without sulfur vacancies and multilayer ZnIn2S4 are all two-dimensional structure of ZnIn2S4. According the reviewers’ suggestion, we have revised the all the ZIS as 2D ZIS in this manuscript.

  1. The results part should start from the discussion on the geometry i.e. combining the XRD, XPS and EPR results. Showing band gap from UV-vis-DRS is too first. However, on the EPR results the from 2D-ZIS and 2D-ZIS-Svare markedly different than the literatures Refs. Journal of Energy Chemistry 58, 408-414 (2021) and ChemCatChem 13, 5148-5155 (2021). Thus, additional discussion is required by keeping in mind the difference of the 2D and bulk nature of the ZIS.

Reply: Thanks for the reviewer’s suggestions. We have adjusted the discussion of UV-vis DRS after EPR and XPS (Fig. 3). We have adjusted the direction of the EPR horizontal axis from small to large. We increased the additional discussion between 2D ZIS, 2D ZIS-Sv and multilayer 2D ZIS.

The high-resolution XPS spectra of S 2p in all specimens can be fitted to two peaks that are assigned to divalent sulfide ions (S2). The binding energies of S 2p3/2 (161.63 eV) and S 2p1/2 (162.88 eV) in 2D ZIS-Sv are negative and broader than those of 2D ZIS (161.90 eV and 163.04 eV) and multilayer 2D ZIS (161.94 eV and 163.09 eV), indicating that sulfur vacancies are existed because of low-coordination sulfur [27]. To further prove the existence of sulfur vacancies in the samples, EPR was measured, as shown in Fig. 3A. An obvious resonance signal at g=2.003 was detected only in 2D ZIS-Sv and this signal was resulted from the existence of unpaired electrons of vacancies [28]. Through above analysis, it can be inferred that abundant sulfur vacancies are existed in 2D ZIS-Sv.

Fig. 2 High-resolution XPS spectra for Zn (A), In (B) and S(C) of 2D ZIS, 2D ZIS-Sv and multilayer 2D ZIS.

Fig. 3 EPR and UV-vis DRS of 2D ZIS, 2D ZIS-Sv, and multilayer 2D ZIS.

  1. Tauc plotted on DRS for band gap values must be correlated with the PL data, which could be the next subsection, while I do not see very clear correlation. Thus, common reader would have difficulty to understand the band gap evolution in few and multi layers of 2D-ZIS and or 2D-ZIS-Vs

Reply: Thanks for the reviewer’s suggestions. We have retested the PL of samples and the peak located at ca. 543.6 nm, 545.9 nm and 545.7 nm are appeared in multilayer 2D ZIS, 2D-ZIS-Vs and 2D ZIS. The location of the peaks is correlated with the DRS band gap values. Besides, 2D ZIS-Sv exhibits the weakest PL peak due to the sulfur vacancies effectively inhibiting the recombination of photo-generated carriers and producing more electrons and holes for photocatalysis.

Fig. 7 PL spectra (A) and TRPL decay spectra (B) of samples.

  1. It will much interesting to see the HRTEM even at 5 nm scale (20 nm is too high). Since, vacancies are very prone to present also in the multi-layer ZIS, a rational discussion must be emphasized that how defects and 2D nature of ZIS is important such 2-fold increase of the PEC performance. I do not know if we will call it as a surface S-vacancy or sub-layer vacancies induced surface reconstruction in presence of the transition metals (Zn) ? This must be justified.

Reply: We support a higher resolution of HRTEM at 5 nm scale in Fig. 1B. Besides, the defects and vacancies can be observed in inset of Fig. 1B. The introduced sulfur vacancies in 2D ZIS serve as electron trapping centers which favor to transfer the electrons form CB to sulfur vacancies, and benefit to the separation of electrons and holes. Besides, sulfur vacancies influence the electrons density of 2D ZIS and elevate the position of CB, allowing photo-excited electrons with stronger reducing ability to take part in the photoelectrochemical process. We believe sulfur vacancies are the main factor for improving the performance, and these defects and vacancies can be observed in inset of Fig. 1B. Xu et. al. reported that Sv can be divided into bulk Sv and surface Sv according to their spatial location. Bulk sulfur vacancies (Sv) enhance carrier separation efficiency of ZnIn2S4. Appropriate surface Sv is helpful for surface carrier transfer, while excessive surface Sv will act as a carrier recombination center which is not conducive to surface carrier injection (Adv. Energy Mater. 2021, 11, 2101181). Combined with the reported literatures (J. Mater. Sci. 2021, 56, 19439-19451; Mater. Sci. Semicond. Process. 2022, 143, 106547), we believe that surface S-vacancy and sub-layer vacancies are all existed in ZnIn2S4. The reviewer gives a significant enlightenment to identify the special role of surface Sv and sub-layer vacancies in the further study.

Reviewer 2 Report

This manuscript describes an original approach consisting in playing with vacancies in ultrathin 2D ZnIn2S4 nanosheets to increase the photoelectrochemical activity. Generally speaking the study is well-driven and the experimental results support the conclusions. Consequently, the reviewer recommends the publication of this manuscript after minor revisions according to the following comments:

  1. The resolution of the XRD patterns of ZIS, ZIS-Sv and multilayer ZIS depicted in Figure 1 is low. Moreover, the diffraction line are large and a broad peak is observed at low angles which suggests the presence of amorphous phase. As a result the authors should show XRD patterns with a better resolution, should quantify the amorphous phase and comment about its nature.
  2. SEM and TEM images shown Figure 1 are not consistent with the preparation of nanosheets. Could the authors record and show more convincing images?
  3. What is the specific area of the different photoelectrode studied? Could the difference in photoelectrochemical activity be due to specific surface area variation?
  4. Interestingly, the authors have used XPS data and UV-visible spectroscopy data to estimate the enrgy level of the valence and conduction bands (Figure 8). However, the method used to establish these energies is not correct. Indeed, in Figure 8 the XPS data clearly show that the Evb-Efermi difference is 1.76, 1.30 and 1.67 eV for ZIS, ZIS-Sv and multilayer ZIS, respectively. However, Figure 9 suggests that Efermi is located at the same energy level as the standard hydrogen electrode: could the author justify this assumption? For the reviewer, the right method to determine the band diagram of such a system is to determine the work function (Fermi level) of the sample by Ultra-violet spectroscopy (UPS) then deduce the valence band-edge energy using the Evb-Efermi difference and finally, use the optical band gap or even better IPES data, to evaluate the energy level of the conduction band edge. This point should be clarify before publication.

Author Response

Reviewer2

This manuscript describes an original approach consisting in playing with vacancies in ultrathin 2D ZnIn2S4 nanosheets to increase the photoelectrochemical activity. Generally speaking the study is well-driven and the experimental results support the conclusions. Consequently, the reviewer recommends the publication of this manuscript after minor revisions according to the following comments:

  1. The resolution of the XRD patterns of ZIS, ZIS-Sv and multilayer ZIS depicted in Figure 1 is low. Moreover, the diffraction line are large and a broad peak is observed at low angles which suggests the presence of amorphous phase. As a result the authors should show XRD patterns with a better resolution, should quantify the amorphous phase and comment about its nature.

Reply: Thanks for the reviewer’s suggestions. We retested the XRD patterns of ZIS, ZIS-Sv and multilayer ZIS at 2°/min scanning rate. The broad peak observed at 6.5° is assigned to ZnIn2S4 (JCPDS 65-6023).

Fig. 1 XRD patterns (A), SEM (D and G), HRTEM (B), and the STEM (C), elemental mappings (E) of Zn (F), In (H) and S (I) of 2D ZIS-Sv.

  1. SEM and TEM images shown Figure 1 are not consistent with the preparation of nanosheets. Could the authors record and show more convincing images?

Reply: Thanks for the reviewer’s suggestions. We have provided one more SEM image in Fig. 1 (Fig. 1D) for the reader to better observe the nanosheets.

  1. What is the specific area of the different photoelectrode studied? Could the difference in photoelectrochemical activity be due to specific surface area variation?

Reply: All the (photo)-electrochemical tests are tested by one L-type glassy carbon electrode (effective area: 0.196 cm2). This eliminates the difference in performance caused by different specific surface area.

  1. Interestingly, the authors have used XPS data and UV-visible spectroscopy data to estimate the enrgy level of the valence and conduction bands (Figure 8). However, the method used to establish these energies is not correct. Indeed, in Figure 8 the XPS data clearly show that the Evb-Efermidifference is 1.76, 1.30 and 1.67 eV for ZIS, ZIS-Sv and multilayer ZIS, respectively. However, Figure 9 suggests that Efermi is located at the same energy level as the standard hydrogen electrode: could the author justify this assumption? For the reviewer, the right method to determine the band diagram of such a system is to determine the work function (Fermi level) of the sample by Ultra-violet spectroscopy (UPS) then deduce the valence band-edge energy using the Evb-Efermi difference and finally, use the optical band gap or even better IPES data, to evaluate the energy level of the conduction band edge. This point should be clarify before publication.

Reply: Thanks for the reviewer’s suggestions. All the energy levels are corresponding to the standard hydrogen electrode. We have corrected the calculation of Evb and the revised results as follows. We use VB XPS spectra to obtain the valence band (VB) potentials of 2D ZIS, 2D ZIS-Sv and multilayer 2D ZIS (Fig. 8). 1.51, 1.14, and 1.45 eV (vs. NHE) were estimated for 2D ZIS, 2D ZIS-Sv and multilayer 2D ZIS, respectively (According formula EVB, NHE = φ + EVB, XPS - 4.44, Nano Energy 2021, 81, 105671). The conductor band (CB) potentials were estimated to be -0.69, -0.98, and -0.75 eV for 2D ZIS, 2D ZIS-Sv and multilayer 2D ZIS, respectively.

Fig. 8 XPS valences of 2D ZIS (A), 2D ZIS-Sv (B) and multilayer 2D ZIS (C).

Round 2

Reviewer 1 Report

Dated: 28-03-2022

Dear Editor, dear Authors, many the updated manuscript. Though there is a substantial improvement of the results, but some minor issues are still required to answer.

The band gap of the various 2D nanosheets of the ZIS herein is about the same 2.1-2.2 eV thus in terms of the photon absorption they are invisibly different while only the S vacancy and redox position of the bands are critical aspects, follows by the relative time difference of the recombination of carriers (nearly 5 ns vs. 7 ns in vacancy vs. cleans models)

Now the concern is on the recombination mediated through the S vacancy models, kindly note that referring the Ref. 27 is not directly linked herein since the authors in this literature are using the Zn vacancy at the poor and rich conditions in bulk, which is in contrast to the current S vacancy model. So please take time to realize these differences and discuss the superior nature of the 2D ZIS.

In the XPS spectra we have would have Supporting figure for the multilayer ZIS as reference to the Ref. 27 by comparison with major focus would be on the S vacancy XPS for 2D-ZIS-Vs, because this is what one of the novelties is discovered herein as supported with the EPR. More justification is required why such small negative shift (0.2-03 eV) is possibly linked to the S vacancies and novel herein.

Finally, the redox positions of these 3 compounds of the 1.4 eV vs 1.7 eV respectively in the S vacancy model and pristine nanosheets vs. NHE or SHE need more justifications, because a it is related to the which type of the PEC reactions or Photocurrent measurements are done. Only sentences though the line 182-183 are too weak at this stage.

Some minor issue, for example the arrangement of the panel in the Fig. 2 or 8 needs to be similar to the Fig. 5

I will update the Fig. 1, with a schematic figures of the 2D nanosheets by replacing the panels D to I by moving them to the Supporting information.

Author Response

Response to Reviewer

We are thankful to the constructive comments from reviewer and we have revised the manuscript according to the comments point by point.

Dear Editor, dear Authors, many the updated manuscript. Though there is a substantial improvement of the results, but some minor issues are still required to answer.

The band gap of the various 2D nanosheets of the ZIS herein is about the same 2.1-2.2 eV thus in terms of the photon absorption they are invisibly different while only the S vacancy and redox position of the bands are critical aspects, follows by the relative time difference of the recombination of carriers (nearly 5 ns vs. 7 ns in vacancy vs. cleans models)

Reply: Thanks for the reviewer’s suggestion. The band gap size of various 2D nanosheets of the ZIS is similar. The introduction of sulfur vacancies mainly changes the electrons structure that decreases the recombination of photocarriers.

Now the concern is on the recombination mediated through the S vacancy models, kindly note that referring the Ref. 27 is not directly linked herein since the authors in this literature are using the Zn vacancy at the poor and rich conditions in bulk, which is in contrast to the current S vacancy model. So please take time to realize these differences and discuss the superior nature of the 2D ZIS. In the XPS spectra we have would have Supporting figure for the multilayer ZIS as reference to the Ref. 27 by comparison with major focus would be on the S vacancy XPS for 2D-ZIS-Vs, because this is what one of the novelties is discovered herein as supported with the EPR. More justification is required why such small negative shift (0.2-03 eV) is possibly linked to the S vacancies and novel herein.

 Reply: Thanks for the reviewer’s suggestion. We have re-referenced the Ref. 19 (Half-unit-cell ZnIn2S4 monolayer with sulfur vacancies for photocatalytic hydrogen evolution. Appl. Catal., B 2019, 248, 193-201) which is S vacancy model of 2D ZIS. The binding energies of S 2p3/2 (161.63 eV) and S 2p1/2 (162.88 eV) in 2D ZIS-Sv are negative and broader than those of 2D ZIS (161.90 eV and 163.04 eV) and multilayer 2D ZIS (161.94 eV and 163.09 eV), indicating that sulfur vacancies are existed because of low-coordination sulfur (Appl. Catal., B 2019, 248, 193-201). The binding energies of Zn 2p3/2 (1021.56 eV) and Zn 2p1/2 (1044.71 eV) are negative shifted ca. 0.40 eV than multilayer 2D ZIS (1021.99 eV and 1045.07 eV), however, this phenomenon is not obvious by In 3d, suggesting that the loss of sulfur atoms around Zn atoms instead of In atoms (Appl. Catal., B 2019, 248, 193-201).

Fig. 2 High-resolution XPS spectra for Zn (A), In (B) and S(C) of 2D ZIS, 2D ZIS-Sv and multilayer 2D ZIS.

Finally, the redox positions of these 3 compounds of the 1.4 eV vs 1.7 eV respectively in the S vacancy model and pristine nanosheets vs. NHE or SHE need more justifications, because a it is related to the which type of the PEC reactions or Photocurrent measurements are done. Only sentences though the line 182-183 are too weak at this stage.

Reply: Thanks for the reviewer’s suggestion. We have enriched the explanation of mechanism of photoelectrochemical water splitting as follows: A possible mechanism based on the band structures is proposed in Fig. 9. It is clearly see that the sulfur vacancies change the CB and VB potentials and the changed CB potential are more negative than others and H+/H2 (0 eV). Sulfur vacancies influence the electrons density of 2D ZIS and elevate the position of CB, allowing photo-excited electrons with stronger reducing ability to take part in the photoelectrochemical process. Besides, due to the reduction of thickness in 2D ZIS, the photogenerated electrons and holes can faster, effectively and barrier-less reach the material surface and take part in redox reactions. The introduced sulfur vacancies in 2D ZIS serve as electron trapping centers which favor to transfer the electrons form CB to sulfur vacancies, and benefit to the separation of electrons and holes. After that, the separated electrons reduce H+ to H2 in cathode and the holes oxidize H2O to O2 in anode.

Some minor issue, for example the arrangement of the panel in the Fig. 2 or 8 needs to be similar to the Fig. 5. I will update the Fig. 1, with a schematic figures of the 2D nanosheets by replacing the panels D to I by moving them to the Supporting information.

Reply: Thanks for the reviewer’s suggestion. Fig. 5 is the EIS of samples, and herein, we focus on the effect of light irradiation on the interfacial electron transfer. Therefore, we separate the drawing the samples’ EIS for better observing the different of resistance under light and in dark. Fig.2 is the XPS spectra of samples. Herein, in order to compare the shift of S, Zn and In elements clearly between samples, we adopted this arrangement of panel. Fig. 8 is XPS valences of samples. The arrangement of the samples’ XPS valence in three graphs is better for observing the curves and get the intersection points. Besides, Fig. 1D-1I is the mapping elemental mappings (E) of Zn (F), In (H) and S (I) of 2D ZIS-Sv. These images show the significant element distribution in 2D ZIS-Sv and we hope to put them in the main manuscript.
